# Influence of the Pecking Motion Frequency on the Cyclic Fatigue Resistance of Endodontic Rotary Files

**DOI:** 10.3390/jcm9010045

**Published:** 2019-12-24

**Authors:** Álvaro Zubizarreta-Macho, Jesús Mena Álvarez, Alberto Albaladejo Martínez, Juan José Segura-Egea, Javier Caviedes Brucheli, Rubén Agustín-Panadero, Roberto López Píriz, Óscar Alonso-Ezpeleta

**Affiliations:** 1Department of Endodontics, Faculty of Health Sciences, Alfonso X el Sabio University, 28691 Madrid, Spain; jmenaalvarez@gmail.com; 2Department of Dentistry, School of Medicine, University of Salamanca, 37008 Salamanca, Spain; albertoalbaladejo@hotmail.com; 3Faculty of Dentistry, University of Sevilla, C/Avicena s/n, 41009 Sevilla, Spain; segurajj@us.es; 4Centro de Investigaciones Odontológicas (CIO) Pontificia, Universidad Javeriana, Bogotá 1101, Colombia; javiercaviedes@gmail.com; 5Department of Stomatology, Faculty of Medicine and Dentistry, University of Valencia, 46010 Valencia, Spain; rubenagustinpanadero@gmail.com; 6Institute of Materials Science of Madrid, Superior Council of Scientific Investigations, 28222 Madrid, Spain; lopezpiriz@gmail.com; 7Department of Endodontics, School of Health Sciences, University of Zaragoza, 22006 Aragorn, Spain; lalonezp@unizar.es

**Keywords:** endodontics, cyclic fatigue, pecking motion, endodontic rotary files, NiTi rotary files

## Abstract

Purpose: To analyze the influence of the pecking motion frequency on the cyclic fatigue resistance of endodontic rotary files. Material and Methods: Sixty PlexV 25.06 endodontic rotary files were selected and distributed into three groups: 30 movements/min (*n* = 20), 60 movements/min (*n* = 20), and 120 movements/min (*n* = 20). A dynamic cyclic fatigue device was designed using Computer Aided Design/ Computer Aided Engineering (CAD/CAE) technology and manufactured by 3D impressions to simulate the pecking motion performed by an operator. Failures of the endodontic rotary files were detected by a Light-Emitting Diode (LED)/Light-Dependent Resistor (LDR) system controlled by an Arduino-Driver complex and management software. Endodontic rotary files were tested on an artificial root canal manufactured by wire electrical discharge machining (EDM), with similar dimensions to those of the instrument under examination. Endodontic rotary files were used following the manufacturer’s recommendations. The results were analyzed by ANOVA and Weibull statistics. Results: All pairwise comparisons revealed statistically significant differences in all three variables, except for the difference in the number of cycles between the groups with 60 and 120 movements/min (*p* = 0.298). The scale distribution parameter of Weibull statistics showed statistically significant differences in all three variables, except for the differences in the number of cycles between groups with 30 and 60 movements/min (*p* = 0.0722). No statistically significant differences in the three variables were observed for the shape distribution parameter. Conclusion: A low frequency of pecking motion is recommended to reduce the risk of failure of endodontic rotary files associated with cyclic fatigue.

## 1. Introduction

Endodontic rotary files have experienced continuous development since nickel–titanium (NiTi) files were introduced in the 1980s [1]. This alloy increases the flexibility and strength of rotary files compared with stainless-steel instruments [2], and it simplifies the endodontic procedure by improving the speed, accuracy, and safety of root canal shaping [3]. Despite continuous enhancements in the design and manufacture of NiTi rotary files to reduce the occurrence of failure during root canal shaping [4], failures can still occur. Many variables can contribute to file separation, but the main causes are cyclic bending fatigue and torsional overload [5,6,7,8]. Torsional overload is caused by the blockage of the endodontic files during rotational movement [9]. However, NiTi instrument failures are primarily caused by cyclic fatigue, which occurs when a NiTi endodontic instrument rotates in a curved root canal [10]. During rotation, the structure of the endodontic instrument is alternately subjected to compressive and tensile stress cycles, which produce microstructural changes that lead to the failure of the endodontic rotary file [11]. Root canal shape, instrument geometry, rotational speed, torque, instrument surface treatments, sterilization cycles, the number of clinical uses, and the chemical composition of NiTi alloys are the main factors that affect the number of cycles to failure (NCF) of NiTi rotary instruments [12,13]. However, the influence of the pecking motion (frequency of in-and-out movement) on the cyclic fatigue resistance of endodontic rotary instruments has never been tested. In 2002, the American National Standard Institute and the American Dental Association standardized a protocol for testing the torsional and flexibility resistance of stainless-steel manual files [14], which was also addressed in the 3630-1 norm by the International Organization for Standardization (ISO) [15] for endodontic instruments with a taper of 2%. However, there is no international standard for testing the cyclic fatigue behavior of NiTi endodontic rotary instruments [11], and several self-designed devices and methods have been used [11]. However, none of these custom-made devices have been capable of dynamically testing the cyclic fatigue of NiTi endodontic rotary instruments in vitro with an automatic detection system and an anatomically based artificial root canal.

The aim of this work was to analyze the influence of the pecking motion on the cyclic fatigue resistance of endodontic rotary instruments; the null hypothesis (H0) asserts that the frequency of the pecking motion does not significantly affect the time to failure, the NCF, or the cyclic fatigue resistance of endodontic rotary instruments.

## 2. Materials and Methods

### 2.1. Study Design

Sixty sterile NiTi CM Wire endodontic rotary instruments 25.06 (Plex V2^®^, Orodeka, Italia) were involved in this study. Before the experiment, every endodontic rotary instrument was inspected for defects or deformities under a stereomicroscope (SZR-10, Optika, Bergamo, Italy), and none were discarded. A randomized controlled experimental trial was performed at the Dental Centre of Innovation and Advanced Specialties at the Alfonso X el Sabio University (Madrid, Spain) between February and July 2019. The endodontic rotary instruments were randomized (Epidat 4.1, Galicia, Spain) and distributed into the three groups: (A) 30 pecking movements/min (*n* = 20), (B) 60 pecking movements/min (*n* = 20), and (C) 120 pecking movements/min (*n* = 20).

### 2.2. The Experimental Cyclic Fatigue Model

The cyclic fatigue tests were performed using a custom-made device (Utility Model Patent number ES1219520) that provides information about the behavior of an endodontic rotary file during a root canal treatment. The endodontic rotary instruments were neither used nor submitted to sterilization cycles before the tests. The structure was designed by 2D/3D Computer-Aided Design/Computer-Aided Engineering (CAD/CAE) (Midas FX+^®^, Brunleys, MK, UK) and manufactured by 3D impression (ProJet^®^ 6000. 3D Systems©, Rock Hill, SC, USA) (Figure 1).

The endodontic handpiece (X-Smart Plus, Dentsply Maillefer, Baillagues, Switzerland) was scanned (3D Geomagic Capture Wrap, 3D Systems©, Rock Hill, SC, USA) to create an accurate design for its support piece by means of inverse engineering technology (Midas FX+^®^, Brunleys) and 3D impression manufacture (ProJet^®^ 6000. 3D Systems©, Rock Hill, SC, USA). This support was firmly attached to the main structure and enabled the angular displacement of the endodontic handpiece by means of a spindle to test different file lengths. The support also allowed the files to be removed during the cyclic fatigue tests.

The direction and speed of the movement were produced by the brushed DC gearmotor (Ref.: 1589, Pololu^®^ Corporation, Las Vegas, NV, USA) controlled by the driver (Ref.: DRV8835, Pololu^®^ Corporation, Las Vegas, NV, USA), which performed an H-bridge function that controlled the speed movement through Pulse Width Modulation (PWM) signals emitted by four switches modulated by transistors. The movement was transferred to the artificial root canal support through a roller bearing system (Ref.: MR104ZZ, FAG, Schaeffler Herzogenaurach, Germany). The artificial root canal support moved in a pure axial motion through a lineal guide (Ref.: HGH35C 10249-1 001 MA, HIWIN Technologies Corp., Taichung, Taiwan). The endodontic rotary file selected (Plex V2^®^, Orodeka) was digitized using a micro Computer Tomography (Skyscan 1176, Bruker-MicroCT, Kontich, Belgium) to obtain a stereolithography (STL) file (Figure 2A) that was needed to design an accurate artificial root canal regarding the measurements of the endodontic rotary file tested (Figure 2B). The artificial root canal piece was manufactured with stainless steel with a 1 mm width. The artificial root canal form was anatomically based and designed using 2D/3D CAD/CAE software (Midas FX+^®^, Brunleys), and it was manufactured by electrical discharge machining (EDM) molybdenum wire-cut technology (Cocchiola S.A., Buenos Aires, Argentina) (Figure 2C) to simulate the tested endodontic rotary instrument’s apical size, taper, and length and enable intimate contact between the endodontic rotary file and the artificial root canal, with a 60° curvature according to Schneider’s measure technique [16] and a 3 mm radius of curvature (Figure 2D).

The failure of the endodontic rotary instrument (Figure 2B) was detected through a Light-Dependent Resistor (LDR) sensor (Ref.: C000025, Arduino LLC^®^, Ivrea, Italy) located at the apex of the artificial root canal. The LDR sensor quantifies the continuous light source emitted by a high-brightness white Light-Emitting Diode (LED) (20000 mcd) (Ref.: 12.675/5/b/c/20k, Batuled, Coslada, Spain), which is opposite to the artificial root canal (Figure 1). The LDR (Ref.: C000025, Arduino LLC^®^) sensor data were conditioned by a processor (Arduino UNO Rev. 3, Arduino LLC^®^, Ivrea, Italy) (Figure 1) to detect values from 0 (endodontic rotary instrument inside the artificial root canal) to 1024 (endodontic rotary instrument outside the artificial root canal). The time to failure was determined when the LDR (Ref.: C000025, Arduino LLC^®^) sensor detected no variations in light values for 50 ms. The hardware was managed by software that receives input signals from the Arduino board (Figure 3A–C). The signals were detected by the LDR (Ref.: C000025, Arduino LLC^®^) sensor with a frequency of 50 ms to accurately detect the time of failure. 

Once the LDR (Ref.: C000025, Arduino LLC^®^) sensor detects the failure of the endodontic rotary instrument, the brushed DC gearmotor stops immediately, and the time to failure and the test parameters are saved by the management software. In addition, the manager application sends output data that start each cyclic fatigue test and control the speed of the pecking motion of the artificial root canal. The speed of the movement and the LDR (Ref.: C000025, Arduino LLC^®^) sensor values were also shown in real time on a Liquid Crystal Display (LCD) (Ref.: LCD-09568, Spark Fun Electronics, Niwot, CO, USA) placed on the structure of the device (Figure 1).

The endodontic rotary instruments were operated by a 6:1 reduction handpiece (X-Smart Plus, Dentsply Maillefer) and a torque-controlled motor with continuous rotation at 400 rpm and 3.5 N/cm torque according to the manufacturer’s instructions. The friction between the file and the artificial canal walls was reduced by applying special high-flow synthetic oil designed for the lubrication of mechanical parts (Singer All-Purpose Oil; Singer Sewing Company, Barcelona, Spain).

All endodontic rotary instruments were rotated until fracture occurred. The number of cycles to fracture (NCF) for each instrument was calculated by using the following formula: NCF = time (seconds) to failure × rotational speed (rpm)/60 seconds [17]. The time to failure, the NCF, the number of in-and-out movements, and the length of the fracture file tip were also measured and recorded. Fractographic analysis of the failure was performed under a scanning electron microscope (ZEISS Supra 35VP; Oberkochen, GmBH, Germany) to examine topographic features of the fractured endodontic rotary files.

### 2.3. Statistical Tests

Statistical analysis of all variables was carried out using SAS 9.4 (SAS Institute Inc., Cary, NC, USA). Descriptive statistics are expressed as means and standard deviations (SD) for quantitative variables and as absolute numbers and percentages for qualitative variables. Comparative analysis was performed by comparing the time to failure, the NCF, and the number of pecking movements (cycles of in-and-out movements) using ANOVA. In addition, Weibull characteristic strength (σ0) and Weibull modulus (m) were calculated. The statistical significance was set at *p* ˂ 0.05.

## 3. Results

The means and SD values for the time to failure (seconds), the NCF, and the number of cycles of in-and-out movement in the study groups are displayed in Table 1, Table 2 and Table 3, respectively. 

The ANOVA revealed statistically significant differences in all three variables. The differences revealed by pairwise comparisons were all statistically significant, except for the difference in the number of cycles between the groups with 60 and 120 movements/min (*p* = 0.298). The scale distribution parameter (η) of Weibull statistics showed statistically significant differences in all three variables, except for the difference in the number of cycles between the groups with 30 and 60 movements/min (*p* = 0.0722) (Table 4, Table 5 and Table 6).

However, the shape distribution parameter (β) showed no statistically significant differences in the time to failure between the groups with 30 and 60 movements/min (*p* = 0.0911), 30 and 120 movements/min (*p* = 0.1537), and 60 and 120 movements/min (*p* = 0.8568) (Figure 4 and Figure 5).

There were no statistically significant differences in the β value of the NCF between the groups with 30 and 60 movements/min (*p* = 0.0990), 30 and 120 movements/min (*p* = 0.1537), and 60 and 120 movements/min (*p* = 0.8858) (Figure 6 and Figure 7).

There were no statistically significant differences in the β value of the number of cycles of in-and-out movement between the groups with 30 and 60 movements/min (*p* = 0.0990), 30 and 120 movements/min (*p* = 0.1537), and 60 and 120 movements/min (*p* = 0.8858) (Figure 8 and Figure 9).

The lifespan of the endodontic rotary files submitted to 30 movements/min was higher (430.53 ± 77.71 s than that of instruments submitted to 60 movements/min (235.75 ± 62.55 s) and 120 movements/min (165.20 ± 42.03 s). The mean lengths of the fractured fragments were not statistically significantly different for any of the instruments tested (*p* > 0.05).

## 4. Discussion

The results obtained in the present study lead to the rejection of the null hypothesis (H0), which states that the frequency of pecking motion has no statistically significant effect on the cyclic fatigue resistance of endodontic rotary instruments.

Different causes of fractures of endodontic instruments have been proposed by many authors and include operator experience [18,19], rotational speed [20,21], the number of uses [22], the number of rotations [23], pre-flaring [24], glide path [25], the angle and radius of the curvature [20,26], and the sterilization of instruments [27]. However, the influence of the frequency of pecking motion performed by the operator on the cyclic fatigue resistance of endodontic rotary instruments has not yet been analyzed.

Cyclic fatigue resistance has been experimentally analyzed by using several custom-made devices; however none of them have been capable of testing the cyclic fatigue resistance of NiTi endodontic rotary instruments using an anatomically based artificial root canal that allows intimate contact between the artificial root canal and the endodontic rotary instrument along the file, which might alter the cyclic fatigue resistance of the endodontic rotary instruments and hence the results of the experiment [11]. The dynamic motion produced by a cyclic fatigue test device can be made comparable to the pecking motion performed by the operator during a root canal treatment. The automatic detection system is able to objectively and accurately identify failures of endodontic rotary files.

Dederich and Zakariasen (1986) were first to develop a dynamic testing device, which allowed the endodontic rotary instruments to realize displacement in a pure axial movement [28]. Ray et al. (2007) also performed dynamic cyclic fatigue studies with standardized axial movement, and they concluded that the pecking motion increased the lifespan of the endodontic rotary instruments submitted to cyclic fatigue compared with the results obtained from static cyclic fatigue studies [29]. Hülsmann et al. (2019) reported similar findings to ours and highlighted the differences between static and dynamic cyclic fatigue tests [30]. Nevertheless, only 12% of published cyclic fatigue studies have used a dynamic fatigue device [30]. Most studies that have compared static and dynamic cyclic fatigue studies have concluded that the time to fracture of endodontic rotary instruments submitted to dynamic cyclic fatigue studies was approximately 20–40% higher than that found in static cyclic fatigue studies [26,31,32,33,34,35]. Previous dynamic cyclic fatigue devices have been unable to accurately identify the causes of endodontic rotary instrument failures because of the absence of standardization of the axial direction of the pecking motion. Lateral movements during the pecking motion of an endodontic rotary instrument can lead to a second bending point at the beginning of the artificial root canal, thus distorting the outcome of cyclic fatigue tests. In addition, Plotino et al. (2010) reported the influence of the shape of the artificial root canal on the contact between the endodontic rotary instrument and the artificial root canal. They concluded that artificial root canals must be designed with a focus on the shape of the endodontic rotary file being studied [36]. However, the above-described artificial root canals had a cylindrically shaped circular section that did not allow intimate contact with the endodontic rotary instrument. In an attempt to simulate real clinical conditions, the present study involved an artificial root canal that was designed by using the measurements of the endodontic rotary instrument being studied. 

The higher β values observed for the group with 30 movements/min for all three variables reveal more predictable behavior of the failure of the endodontic rotary instruments in this study group, and the lower η values found in the groups with 60 and 120 movements/min for all three variables indicate a lower cyclic fatigue resistance of the endodontic rotary instruments in these study groups. The location of the crystal structure transformation is changed in the endodontic rotary instrument during the pecking motion, which increases the cyclic fatigue resistance [30].

The conclusion derived from this study is that a high frequency of pecking motion decreases the cyclic fatigue resistance of endodontic rotary files.

Nevertheless, further research is needed to determine the influence of the frequency of pecking motion and working time on the cyclic fatigue resistance of endodontic rotary files.

## 5. Conclusions

In conclusion, within the limitations of this study, our results show that a high frequency of pecking motion decreases the cyclic fatigue resistance of endodontic rotary instruments. A low frequency of pecking motion is recommended to reduce the risk of fractures of endodontic rotary instruments associated with cyclic fatigue.

## Figures and Tables

**Figure 1 jcm-09-00045-f001:**
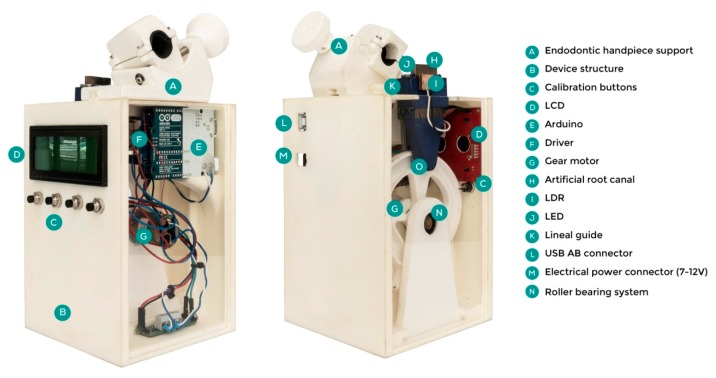
Parts of the hardware of the cyclic fatigue test device.

**Figure 2 jcm-09-00045-f002:**
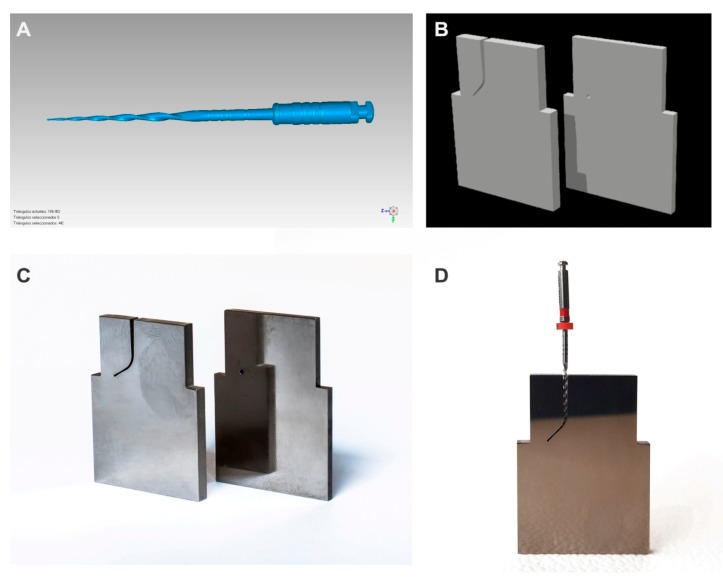
(**A**) Stereolithography (STL) file of the endodontic rotary file; (**B**) STL file of the artificial root canal; (**C**) artificial root canal manufactured by electrical discharge machining (EDM) and (**D**) endodontic rotary file in intimate contact with the artificial root canal (**D**).

**Figure 3 jcm-09-00045-f003:**
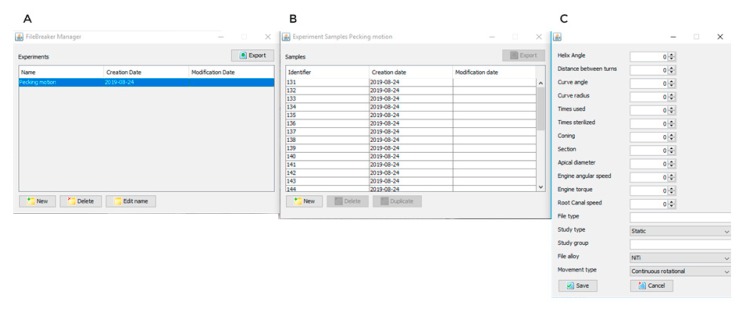
The management software (**A**,**B**) that managed the cyclic fatigue device with configurable parameters (**C**).

**Figure 4 jcm-09-00045-f004:**
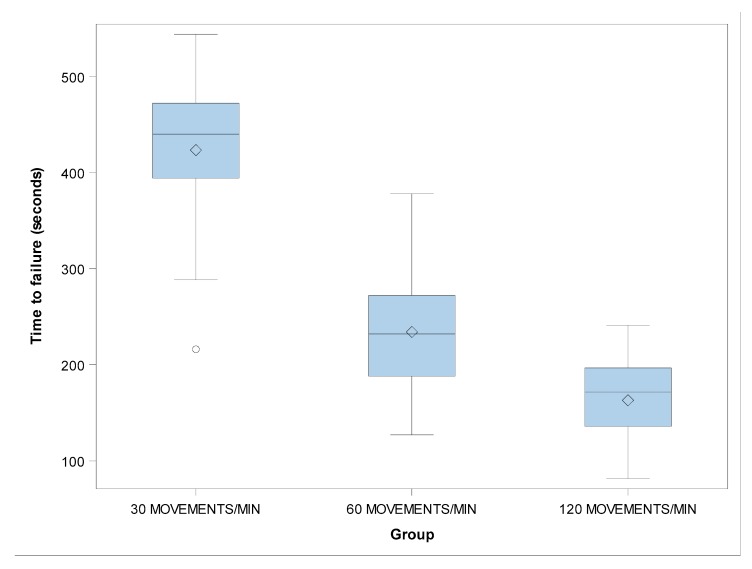
Boxplots of the time to failure of the experimental groups. The horizontal line in each box represents the median value.

**Figure 5 jcm-09-00045-f005:**
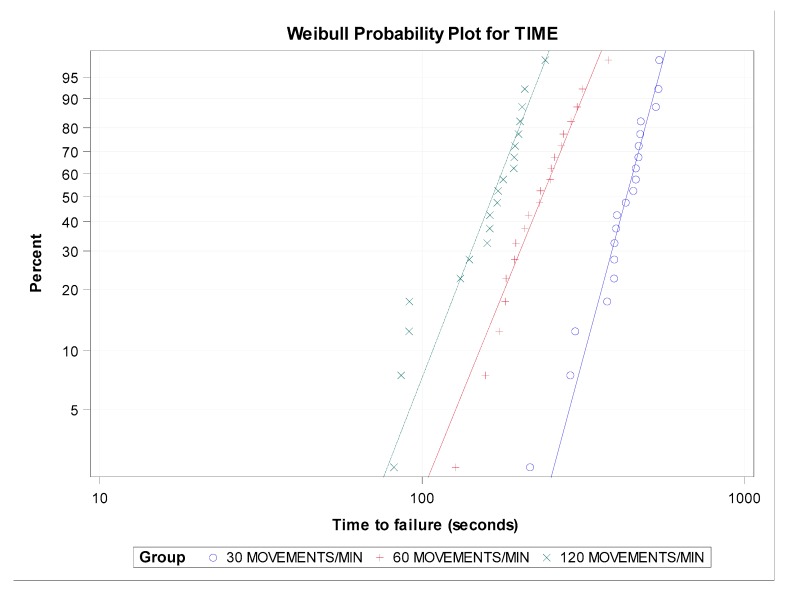
Weibull probability plot of the time to failure.

**Figure 6 jcm-09-00045-f006:**
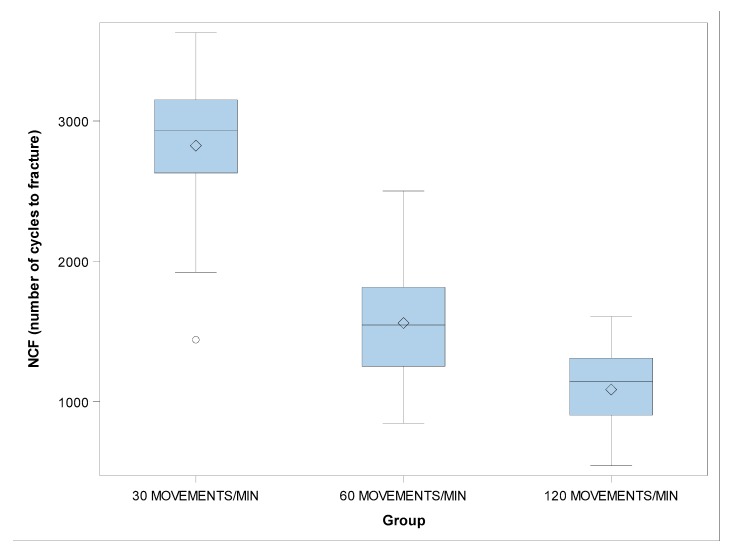
Boxplots of the NCF of the experimental groups. The horizontal line in each box represents the median value.

**Figure 7 jcm-09-00045-f007:**
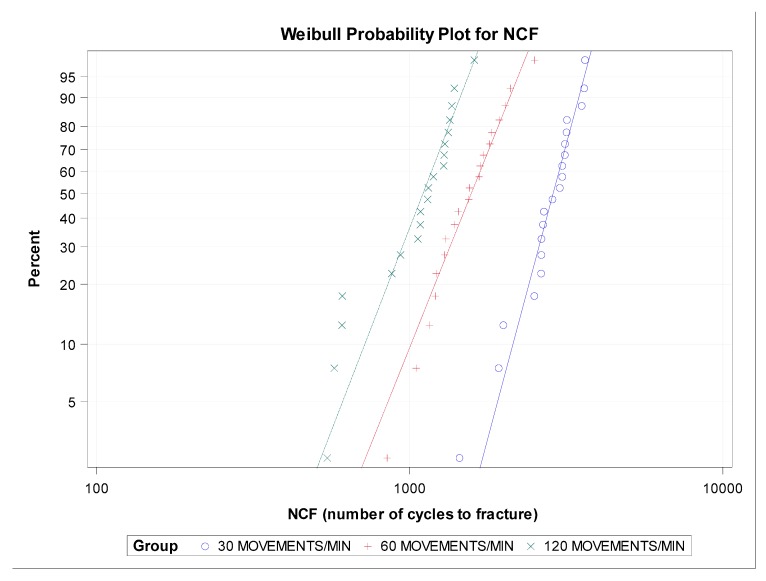
Weibull probability plot of the NCF.

**Figure 8 jcm-09-00045-f008:**
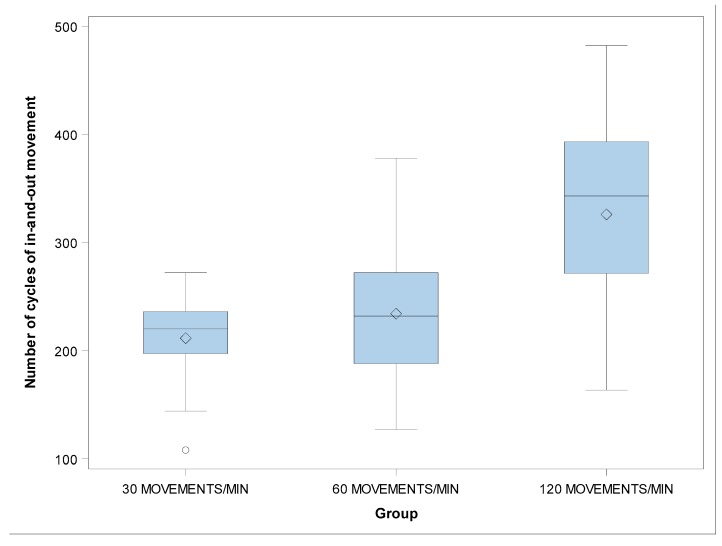
Boxplots of the number of cycles of in-and-out movement of the experimental groups. The horizontal line in each box represents the median value.

**Figure 9 jcm-09-00045-f009:**
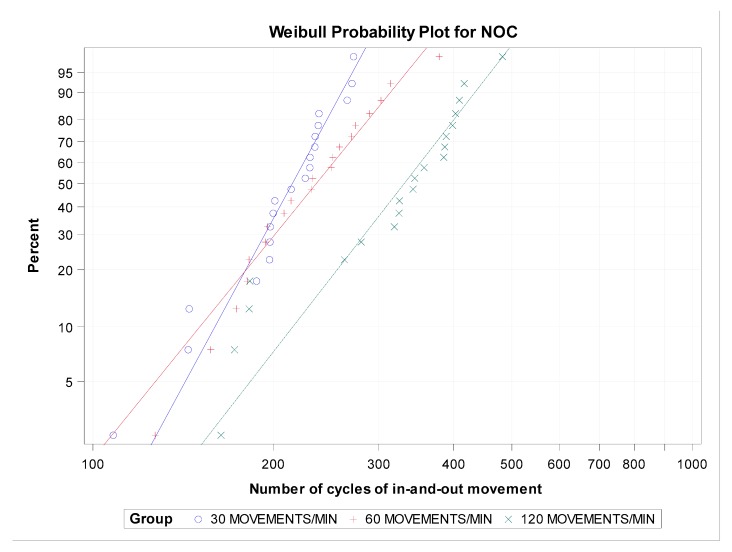
Weibull probability plot of the number of cycles of in-and-out movement.

**Table 1 jcm-09-00045-t001:** Descriptive statistics of the time to failure.

	*n*	Mean	SD	Minimum	Maximum	Fracture Length (mm)
30 MOV/MIN	20	423.66 ^a^	84.61	216.23	544.40	3.23
60 MOV/MIN	20	234.23 ^b^	60.56	127.00	378.21	3.04
120 MOV/MIN	20	163.06 ^c^	45.95	81.80	241.16	3.81

SD, standard deviations; ^a^, ^b^, ^c^, different superscript means statistically significant differences between groups (*p* < 0.05).

**Table 2 jcm-09-00045-t002:** Descriptive statistics of the NCF.

	*n*	Mean	SD	Minimum	Maximum	Fracture Length (mm)
30 MOV/MIN	20	2824.37 ^a^	564.05	1441.55	3629.35	3.23
60 MOV/MIN	20	1560.55 ^b^	401.23	846.69	2501.41	3.04
120 MOV/MIN	20	1087.06 ^c^	306.34	545.33	1607.73	3.81

NCF, the number of cycles to failure; ^a^, ^b^, ^c^, different superscript means statistically significant differences between groups (*p* < 0.05).

**Table 3 jcm-09-00045-t003:** Descriptive statistics of the number of cycles of in-and-out movement.

	*n*	Mean	SD	Minimum	Maximum	Fracture Length (mm)
30 MOV/MIN	20	211.60 ^a^	42.66	108.12	272.20	3.23
60 MOV/MIN	20	234.23 ^b^	60.56	127.00	378.21	3.04
120 MOV/MIN	20	326.12 ^b^	91.90	163.59	482.32	3.81

^a^, ^b^, different superscript means statistically significant differences between groups (*p* < 0.05).

**Table 4 jcm-09-00045-t004:** Weibull statistics of the time to failure.

	m = Weibull Shape (β)				σ_0_ = Weibull Scale (η)			
	Estimate	St Error	Lower	Upper	Estimate	St Error	Lower	Upper
30 MOV/MIN	6.3802	1.1533	4.4768	9.0927	455.9851	16.7468	424.3155	490.0185
60 MOV/MIN	4.2152	0.6990	3.0456	5.8341	257.1689	14.4489	230.3531	287.1065
120 MOV/MIN	4.4090	0.8177	3.0653	6.3417	179.5190	9.5319	161.7759	199.2080

**Table 5 jcm-09-00045-t005:** Weibull statistics of the NCF.

	m = Weibull Shape (β)				σ_0_ = Weibull Scale (η)			
	Estimate	St Error	Lower	Upper	Estimate	St Error	Lower	Upper
30 MOV/MIN	6.3802	1.1533	4.4768	9.0927	3039.9005	111.6454	2828.7697	3266.7895
60 MOV/MIN	4.2540	0.7079	3.0702	5.8944	1712.7561	95.3373	1535.7306	1910.1877
120MOV/MIN	4.4090	0.8177	3.0653	6.3417	1196.7927	63.5463	1078.5059	1328.0528

**Table 6 jcm-09-00045-t006:** Weibull statistics of the number of cycles of in-and-out movement.

	m = Weibull Shape (β)				σ_0_ = Weibull Scale (η)			
	Estimate	St Error	Lower	Upper	Estimate	St Error	Lower	Upper
30 MOV/MIN	6.3342	1.1474	4.4412	9.0341	227.8707	8.4275	211.9375	245.0018
60 MOV/MIN	4.2152	0.6990	3.0456	5.8341	257.1689	14.4489	230.3531	287.1065
120 MOV/MIN	4.4090	0.8177	3.0653	6.3416	359.0376	19.0640	323.5513	398.4160

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
