# Peer review of "Influence of the Pecking Motion Frequency on the Cyclic Fatigue Resistance of Endodontic Rotary Files"

_jcm, 2019, doi:10.3390/jcm9010045_

Round 1

Reviewer 1 Report

This research paper is interesting for the specialists in endodontics. Of course as it is stated by authors further research is needed.

The authors have analysed one of the important aspects during root canal
treatment procedure in experimental study. Of course in clinical situation
You will receive a lot of differences due to the skills of each dentist also
due to clinical environment which will not let us You use pecking movement of
instrument in standartised situation. This article brings information which is important for further research
which could allow us to diminish failures in root canal treatment associated
with the fractures of endodontic instruments. Also it is important evidence
(experimeta) which is important not only for the practising dentists and
endodontists, but also for the educators in dental schools.

Author Response

Dear reviewer 1:

I’m pleased to resubmit the manuscript of the work entitled, “Influence of the pecking motion frequency on the cyclic fatigue resistance of endodontic rotary files

Reviewer 1: I don't feel qualified to judge about the English language and style.

Response: In order to adapt to the reviewer's 1 comments, we have send the manuscript to a specialized traductor.

We take this opportunity to thank the recommendations and suggestions made by the reviewer to improve the document.

Reviewer 2 Report

This article evaluated the effect of pecking frequency on cyclic fatigue resistance. Authors may did effortful work.

Line61-62: there're not new. 3D printing already were used to make artificial canal by 3D printer for metal. Authors do not described details what is “anatomical-based artificial root canal”. When I look into figure and its legend, I can find out any other difference of this model comparing to the models in the previous studies.

Line 64-67: Authors might not search the previous studies of this topic.

The pecking motion are dealt in so many articles including pecking depth, stress generation, cyclic fatigue and root canal preparation quality.

Line 102: Is “Stainless steel” correct?

Line 213-214: it already was analyzed. Please search in “dynamic” cyclic fatigue resistance.

Line220: This article is published 10 years ago. So, lots of things have been updated after that.

Artificial canal also are being made by CAD and electrical discharge machining method.

Authors can find out articles about that.

Line245~:This sentence only discussed about the result of this study. However, There is still lack of discussion of the results in this study.

Author Response

Dear reviewer 2:

I’m pleased to resubmit the manuscript of the work entitled, “Influence of the pecking motion frequency on the cyclic fatigue resistance of endodontic rotary files

Reviewer 2: Line61-62: there're not new. 3D printing already were used to make artificial canal by 3D printer for metal. Authors do not described details what is “anatomical-based artificial root canal”. When I look into figure and its legend, I can find out any other difference of this model comparing to the models in the previous studies.

Response: In order to adapt to the reviewer's 2 suggestions, we have described in the line 104 of the “Material and Methods” section the characteristics of the “anatomical-based artificial root canal”, and we have added a new figure and legend, describing the procedure necessary to obtain this accurate artificial root canal. What makes it different from the artificial root canals used previously is that its morphology is based on the morphology of the endodontic rotary file that will be tested in the study, designed and performed by an accurate digital flow procedure.

Reviewer 2: Line 64-67: Authors might not search the previous studies of this topic. The pecking motion are dealt in so many articles including pecking depth, stress generation, cyclic fatigue and root canal preparation quality.

Response: In order to adapt to the reviewer's 2 suggestions, we would like to highlighted that this novel dynamically custom made cyclic fatigue device, with an automatic detection system, anatomical-based artificial root canal and management software is unique in the world, and proof of this is that the Patent and Trademark Office has granted us the title of "Utility Model Patent number ES1219520” (Line80). If there were any device slightly similar to our cyclic fatigue device, the Patent and Trademark Office would never have given us that title.

Reviewer 2: Line 102: Is “Stainless steel” correct?

Response: In order to adapt to the reviewer's 2 suggestions, we have changed the words.

Reviewer 2: Line 213-214: it already was analyzed. Please search in “dynamic” cyclic fatigue resistance.

Response: In order to adapt to the reviewer's 2 suggestions, we have make another bibliographic search with the key words: “dynamic cyclic fatigue AND pecking motion”, and we have not find any dynamic cyclic fatigue study that analyzed the influence of the frequencies of pecking motion on the cyclic fatigue resistance of endodontic rotary instruments.

Reviewer 2: Line220: This article is published 10 years ago. So, lots of things have been updated after that. Artificial canal also are being made by CAD and electrical discharge machining method. Authors can find out articles about that.

Response: In order to adapt to the reviewer's 2 suggestions, we have make another bibliographic search with the key words: “root canal AND electrical discharge machining”, and we have not find any artificial root canal based in the anatomy of the endodontic rotary file tested and used in a dynamic cyclic fatigue study that analyzed the influence of the frequencies of pecking motion on the cyclic fatigue resistance of endodontic rotary instruments.

Reviewer 2: Line245~:This sentence only discussed about the result of this study. However, There is still lack of discussion of the results in this study.

Response: In order to respond to the reviewer's 2 suggestions, we could not compare our results with another study with a  dynamically custom made cyclic fatigue device, with an automatic detection system, anatomical-based artificial root canal and management software, used in a dynamic cyclic fatigue study that analyzed the influence of the frequencies of pecking motion on the cyclic fatigue resistance of endodontic rotary instruments, because we could not find it and the articles that we could find were too different.

We take this opportunity to thank the recommendations and suggestions made by the reviewers to improve the document.

Reviewer 3 Report

First of all, the document was uploaded on the website with all markups and changes being tracked via MS Word. It made it very difficulty to read. Additionally, the authors' names and institutions were apparent.  I would think that the journal would ensure that these are blinded or anonymous.  The Introduction is good and the Materials and Methods are very thorough. The study is very well designed.  Below are a few comments that I am able to make after treading through the markups and tracked changes.

Line 75 stated 60 sterile rotary instruments were used.  However, in lines 87 and 88 it states the files were neither used or submitted for sterilization.  Please clarify.

Figure 2 has panels A. B, C, D then again with different A and B at the bottom. Please clarify.

What is the model made of that has the simulated canal?

I am not sure if Figure 3 adds anything to the study.

Line 144 is "app" a word officially recognized n English dictionaries?

Table 1 does not have any units of measuring time.  Is this seconds, minutes or hours?  

I am not an expert in Weibull analysis, but when I look at figures 8 and  9 it appears that it requires more cycles of in-and-out movement for failure (for 120 movements/min). It requires more than 400 cycles to reach 100% failure.  But your conclusion states that higher frequency of pecking motion decrease resistance to failure which proposes that it should fail sooner.  Are the graphs labeled wrong or do I just don't understand?

I understand tables 1 and 2,  but explain in the Discussion why more number of cycles (Table 3) leads to  more resistance to fracture.  I takes 326 cycles of 120 mov/min for failure.  Many would think that  30 mov/min would perform better.

Tables 1 through 3 should have super scripts letters indicating which groups are statistically different or the same.

May be Figures 4 and 5 should be presented before Tables 1-3. Or eliminate Tables 1 and 2 as the same information is presented.

Conclusions:  "high frequency of pecking decreases cyclic fatigue, but figures 8 and 9 do not bear this out.  Figures 4 and 5 support the second half your the conclusion that low pecking motion reduces cyclic fatigue.

What are the limitations of this study?

References are fine.

Author Response

Dear reviewer:

I’m pleased to resubmit the manuscript of the work entitled, “Influence of the pecking motion frequency on the cyclic fatigue resistance of endodontic rotary files

Reviewer 3: English language and style are fine/minor spell check required

Response: In order to adapt to the reviewer's 3 comments, we have send the manuscript to a specialized traductor. We attached the Certificate.

Reviewer 3: First of all, the document was uploaded on the website with all markups and changes being tracked via MS Word. It made it very difficulty to read.

Response: In order to adapt to the reviewer's 3 comments, we would like to clarify that this is the sending format requested by the Assistant Editor: "Any revisions should be * clearly highlighted *, for example using the" Track Changes "function in Microsoft Word, so that they are easily visible to the editors and reviewers".

Reviewer 3: Additionally, the authors' names and institutions were apparent. I would think that the journal would ensure that these are blinded or anonymous.

Response: In order to adapt to the reviewer's 3 comments, we would like to clarify that we have adapted to the Submitting Guidelines of the Journal of Clinical Medicine.

Reviewer 3: Line 75 stated 60 sterile rotary instruments were used. However, in lines 87 and 88 it states the files were neither used or submitted for sterilization. Please clarify.

Response: In order to respond to the reviewer's 3 comments, we clarify that each of the instruments had not been used or sterilized before being used for this study. In the lines 88-89 we clarified: “The endodontic rotary instruments were neither used nor submitted to sterilization cycles before the tests”.

Reviewer 3: Figure 2 has panels A. B, C, D then again with different A and B at the bottom. Please clarify.

Response: In order to respond to the reviewer's 3 comments, we clarify that the Figure 2 with 4 panels (A-D) was included Figure 2 at the request of another reviewer, who asked to remove Figure 2 with 2 panels. We have clarify on the manuscript with the "Track Changes" function in Microsoft Word because is the format requested by the Assistant Editor: "Any revisions should be * clearly highlighted *, for example using the" Track Changes "function in Microsoft Word, so that they are easily visible to the editors and reviewers".

Reviewer 3: What is the model made of that has the simulated canal?

Response: In order to respond to the reviewer's 3 comments, we clarify that in the lines 113-114 we wrote: “The artificial root canal piece was manufactured with stainless steel with a 1 mm width”.

Reviewer 3: I am not sure if Figure 3 adds anything to the study.

Response: In order to respond to the reviewer's 3 comments, we clarify that in the lines 137-138 we wrote: “The hardware was managed by software that receives input signals from the Arduino board (Figure 3A, B, C)”. This figure of the management software shows the parameters that can be analyzed by the dynamic cyclic fatigue device.

Reviewer 3: Line 144 is "app" a word officially recognized n English dictionaries?

Response: In order to adapt to the reviewer's 3 suggestions, we have changed the word “app” by the word “application”.

Reviewer 3: Table 1 does not have any units of measuring time. Is this seconds, minutes or hours?

Response: In order to respond to the reviewer's 3 suggestions, we have added the word “seconds” in the line 174.

Reviewer 3: I am not an expert in Weibull analysis, but when I look at figures 8 and  9 it appears that it requires more cycles of in-and-out movement for failure (for 120 movements/min). It requires more than 400 cycles to reach 100% failure. But your conclusion states that higher frequency of pecking motion decrease resistance to failure which proposes that it should fail sooner. Are the graphs labeled wrong or do I just don't understand?

Response: In order to respond to the reviewer's 3 comments, we clarify that the endodontic rotary instruments submitted to 120 mov/min inside the artificial root canal make more pecking movements until their fracture, but they fracture much earlier (Time to Failure) than those submitted to 30 or 60 mov/min. This fact clarifies the sentence of the "Conclusion" section: "In conclusion, within the limitations of this study, our results showed that a high frequency of pecking motion values, results in a further decreases of the cyclic fatigue resistance of endodontic rotary instruments". In addition, the greater slope of groups 120 and 60 mov / min of the Weibull probability plot of Figure 9 indicate a less predictable behaviour of these instruments.

Reviewer 3: I understand tables 1 and 2, but explain in the Discussion why more number of cycles (Table 3) leads to more resistance to fracture. I takes 326 cycles of 120 mov/min for failure. Many would think that 30 mov/min would perform better.

Response: In order to respond to the reviewer's 3 comments, we clarify that the parameter that indicates the resistance to fracture is the "Time to Failure", endodontics want that their endodontic rotary instruments last a long time to failure, but we do not want that the endodontic rotary instruments performed many pecking movements/minute. Imagine that we submitted a endodontic rotary instrument to 1000 mov/second and it lasts 1 second to failure and we submitted an another endodontic rotary instrument to 0,5 mov/sec (30 mov/min) and it lasts 423 seconds to failure. The first one has done move pecking movements, but the second one last more time to fracture.

Reviewer 3: Tables 1 through 3 should have super scripts letters indicating which groups are statistically different or the same.

Response: In order to respond to the reviewer's 3 suggestions, we have added the super scripts letters indicating which groups are statistically different or not.

Reviewer 3: May be Figures 4 and 5 should be presented before Tables 1-3. Or eliminate Tables 1 and 2 as the same information is presented.

Response: In order to adpat to the reviewer's 3 suggestions, we have moved the Figures 4 and 5 before Tables 1, 2 and 3.

Reviewer 3: Conclusions: "high frequency of pecking decreases cyclic fatigue, but figures 8 and 9 do not bear this out. Figures 4 and 5 support the second half your the conclusion that low pecking motion reduces cyclic fatigue.

Response: In order to respond to the reviewer's 3 comments, we clarify that we wrote in the “Conclusion” section: “our results show that a high frequency of pecking motion decreases the cyclic fatigue resistance of endodontic rotary instruments”. We also clarify that the parameter that indicates the resistance to fracture is the "Time to Failure". Figure 8 and 9 represent the Number of Cycles of in-and-out Movements of the endodontic rotary files, 120 mov/min study group performed more pecking movements than 30 and 60 mov/min study groups, but they last more Time to Failure and performed more Number of Cycles to Failure moving on its own axis.

Reviewer 3: What are the limitations of this study?

Response: In order to respond to the reviewer's 3 comments, we clarify that this is an in vitro study and it present limitations as the artificial root canal material that is not dentin. In addition, the endodontics, do not perform a “pure axial motion” during root canal shaping.

We take this opportunity to thank the recommendations and suggestions made by the reviewer to improve the document.

Yours sincerely.

Round 2

Reviewer 2 Report

Authors improved this manuscript. 

Responses to reviewer's comments seem to be proper.

There are some minor typo-errors and grammar problems. It would be improved by further works.

Author Response

Dear reviewer:

I’m pleased to resubmit the manuscript of the work entitled, “Influence of the pecking motion frequency on the cyclic fatigue resistance of endodontic rotary files

Reviewer 2: I don't feel qualified to judge about the English language and style.

Response: In order to adapt to the reviewer's 2 comments, we have send the manuscript to a specialized traductor. We attached the Certificate.

We take this opportunity to thank the recommendations and suggestions made by the reviewer to improve the document.

Yours sincerely